# Risk of Suicidal Behavior in Children and Adolescents Exposed to Maltreatment: The Mediating Role of Borderline Personality Traits and Recent Stressful Life Events

**DOI:** 10.3390/jcm10225293

**Published:** 2021-11-14

**Authors:** Laia Marques-Feixa, Jorge Moya-Higueras, Soledad Romero, Pilar Santamarina-Pérez, Marta Rapado-Castro, Iñaki Zorrilla, María Martín, Eulalia Anglada, María José Lobato, Maite Ramírez, Nerea Moreno, María Mayoral, María Marín-Vila, Bárbara Arias, Lourdes Fañanás

**Affiliations:** 1Department of Evolutionary Biology, Ecology and Environmental Sciences, Faculty of Biology, University of Barcelona, Biomedicine Institute of the University of Barcelona (IBUB), Av Diagonal 643, 2n A, 08028 Barcelona, Spain; laiamarques@ub.edu (L.M.-F.); nereamorenog711@gmail.com (N.M.); barbara.arias@ub.edu (B.A.); 2Network Centre for Biomedical Research in Mental Health (CIBER of Mental Health, CIBERSAM), Av. Monforte de Lemos, 3-5, 28029 Madrid, Spain; jorge.moya@udl.cat (J.M.-H.); sromero@clinic.cat (S.R.); mrapado@iisgm.com (M.R.-C.); inaki.zorrillamartinez@osakidetza.eus (I.Z.); maria.mayoral@iisgm.com (M.M.); 3Department of Psychology, Faculty of Education, Psychology and Social Work, University of Lleida, Av. de I’Estudi General, 4, 25001 Lleida, Spain; 4Department of Child and Adolescent Psychiatry and Psychology, 2017SGR88, Institute of Neuroscience, Hospital Clínic de Barcelona, C/Villarroel, 170, 08036 Barcelona, Spain; psantama@clinic.cat; 5Institut d’Investigacions Biomediques August Pi i Sunyer (IDIBAPS), C/Rosselló, 149, 08036 Barcelona, Spain; 6Department of Child and Adolescent Psychiatry, Institute of Psychiatry and Mental Health, Hospital General Universitario Gregorio Marañón, IiSGM, C. Dr. Esquerdo, 46, 28007 Madrid, Spain; 7Melbourne Neuropsychiatry Centre, Department of Psychiatry, The University of Melbourne & Melbourne Health, C/Alan Gilbert, 161, Carlton, VIC 3053, Australia; 8Department of Psychiatry, Hospital Santiago Apostol, Olagibel Kalea, 29, 01004 Vitoria-Gasteiz, Spain; 9Adolescent Crisis Unit, Hospital Benito Menni, C/Pablo Picasso, 12, 08830 Sant Boi de Llobregat, Spain; mmartin.hbmenni@hospitalarias.es; 10Hospital for Adolescents, Fundació Orienta, c/Sant Lluís, 64, 08850 Gavà, Spain; eanglada@fundacioorienta.com; 11Department of Psychiatry, Puerta de Hierro University Hospital-Majadahonda, C/Joaquín Rodrigo, 1, 28222 Majadahonda, Spain; mjose.lobato@salud.madrid.org (M.J.L.); mmvila@salud.madrid.org (M.M.-V.); 12Galdakao Mental Health Services, Child and Adolescent Mental Health, C/Ibaizabal, 6, 48960 Galdakao, Spain; maite.ramireztrapero@osakidetza.eus

**Keywords:** childhood maltreatment, stressful life event (SLE), borderline personality disorder traits, emotion dysregulation, non-suicidal self-injury (NSSI), suicidal behavior (SB), youth, complex trauma

## Abstract

Childhood maltreatment (CM) is associated with increased non-suicidal self-injury (NSSI) and suicidal behavior (SB), independently of demographic and mental health conditions. Self-Trauma Theory and Linehan’s Biopsychosocial Model might explain the emergence of Borderline Personality Disorder (BPD) symptoms as mediators of the association between CM and the risk of SB. However, little is known regarding such relationships when the exposure is recent for young persons. Here, we study 187 youths aged 7–17, with or without mental disorders. We explore CM experiences (considering the severity and frequency of different forms of neglect and abuse), recent stressful life events (SLEs), some BPD traits (emotion dysregulation, intense anger and impulsivity), and the risk of SB (including NSSI, suicide threat, suicide ideation, suicide plan and suicide attempt). We study the direct and mediating relationships between these variables via a structural equation analysis using the statistical software package EQS. Our findings suggest that youths exposed to more severe/frequent CM have more prominent BPD traits, and are more likely to have experienced recent SLEs. In turn, BPD traits increase the risk of SLEs. However, only emotion dysregulation and recent SLEs were found to be correlated with SB. Therefore, targeted interventions on emotion dysregulation are necessary to prevent NSSI or SB in children and adolescents exposed to CM, as is the minimization of further SLEs.

## 1. Introduction

Childhood is one of the most sensitive and neuroplastic periods of human development, as the stimuli and upbringing experienced during this stage is crucial for the maturation of brain systems and cognitive functions [1,2]. Early adversities, such as childhood maltreatment (CM), can be of detriment to neurodevelopment and can disturb intrapsychic and interpersonal patterns [3,4]. More specifically, when an individual has experienced multiple, severe and pervasive traumatic events during childhood (complex trauma), the psychological outcomes are often also multiple and severe. In this regard, it is not surprising that people with psychiatric disorders and a history of CM represent a clinically distinct subtype of patients, who have a worse clinical prognosis. They are characterized by earlier onset, more severe symptoms and comorbidity, the need for a higher medication dosage, and more frequent and longer hospitalizations [5,6]. Moreover, it seems that the timing, chronicity and the severity of the CM may play a role in clinical outcomes [7,8], establishing a dose–response relationship between multiplicity, severity or frequency of CM exposure on the one hand, and disease outcomes on the other [9,10].

Death by suicide could be one of the most devastating consequences of suffering from CM. Nowadays, suicide is the leading cause of death among young people (15–29 years old) in Spain and the second leading cause of death in Europe [11,12]. Throughout the last year (in the context of COVID-19 pandemic), it seems that suicide attempts began to increase among adolescents aged 12–17 years, especially in girls [13,14]. Although suicide is a multifactorial phenomenon, recent systematic reviews that focused on adolescents [15] and young adults [16] supported the finding that all types of CM are associated with an increased risk of SB. More specifically, Angelakis et al. [16] performed a meta-analysis concluding that complex trauma increased 5-fold the risk of suicide attempts in adults. Moreover, if after their experience of CM, there is an escalation of new stressful life events (SLEs), an increase in suicidal behavior (SB) might emerge [17]. This aspect could be especially relevant in stage in life that is as volatile as adolescence, during which brain regions associated with impulse control are still undergoing development [18].

Experts have also found higher rates of non-suicidal self-injury (NSSI), suicide ideation and other SB among individuals who have suffered CM [16,19]. NSSI is behavior that is not intended to result in the death of the individual and is often related to attempts to temporally alleviate overwhelming negative emotion or to a form of self-directed anger [20]. NSSI and SB can occur in the same individual [21]. In fact, a review on this topic proposes an integrated model with specific testable predictions about this link [22]. Some experts support the idea that NSSI is specifically associated with the transition from suicidal thinking to action in adolescents [23], and it is one of the main predictors of suicide attempts [24]. Thus, it is important to assess a broad spectrum of risk of SB, across the entire range it encompasses, from NSSI, suicide ideation, suicide threat and suicide plans to suicide attempts [25]. 

Nevertheless, there is a major gap in the literature regarding the mechanisms underlying the relationship between CM and SB risk, especially concerning young populations [26,27]. In this regard, a comprehensive theory was provided by Myers and colleagues to provide an understanding of the myriad self-damaging behaviors observed in individuals exposed to CM [28]. This model, called the Self-Trauma Model, proposes that complex trauma results in the maladaptive development of the following three primary capacities: affect regulation, identity, and interpersonal relatedness [29]. Consequently, when the individual with a history of CM is confronted with new stressful circumstances, she or he may be unable to rely on these internal resources and may resort to maladaptive tension-reducing behaviour, such as aggression, substance abuse, risky sexual behaviour, self-injury or SB. Complementarily, Linehan’s Biopsychosocial Model proposed that personality dysfunctions emerge when there is a biological predisposition (impulsivity followed by heightened emotional sensitivity) combined with an emotionally invalidating form of caregiving, CM being one of the most severe forms of emotional invalidation [30]. From a neuropsychological perspective of suicide risk, Allen et al. [31] proposed emotional dysregulation as a “multi-final common pathway” through which disparate diatheses (including CM) operate to influence varied adverse clinical outcomes.

In fact, some studies in an adult population have already described that the Self-Trauma Model provides support for the emergence of Borderline Personality Disorder (BPD) symptoms (affective dysregulation, identity problems or paranoia), a significant mediator of this relationship between CM and risk of SB [32]. Moreover, there seems to be a dose–effect relationship whereby the greater the exposure to early adversities, the more severe the Personality Disorders (PDs) [33], with increasing disturbances in the functioning of aspects of the self and interpersonal dysfunction across various contexts and relationships, such as a high risk of NSSI or SB. Therefore, considering that individuals with BPD are almost 14 times more likely to report a history of CM [27] and present higher rates of NSSI and SB [34,35], it would be interesting to consider, especially in the young population, BPD traits as possible mediators between CM and the risk of SB [30]. 

The diagnosis of PDs in childhood and in adolescence remains controversial [36,37] and many clinicians are reluctant to apply such diagnoses in younger individuals [38]. Nevertheless, previous research demonstrates that a considerable proportion of individuals with BPD traits prior to the age of 19 continue to manifest those symptoms for up to 20 years (from 14% to 40%) and that such traits at a young age predict long-term deficits in functioning [39]. Furthermore, considering that personality traits that increase the likelihood to risky behaviours are found in the general population, especially during adolescence, the use of a continuum of PD traits could be more effective. For instance, some authors support the claim that the traits most frequently exhibited by those who meet the criteria for BPD are emotion dysregulation, intense anger, impulsivity, and indirect aggression [40,41,42]. Along these themes, the literature shows that CM leads to experiences of chronic emotion dysregulation that might provide the basis of impairment and further exposure to trauma, as well as potentiating NSSI [43] and suicide ideation and suicide attempts [44]. In fact, emotion dysregulation is widely reported to be a transdiagnostic link between CM with general psychopathology [45,46]. 

To summarise, prior research focused on the mediating role of BPD traits in the association between CM and risk of SB, rarely included children and adolescents. In this study, we are particularly interested in elucidating the relationship between some BPD traits (emotion dysregulation, intense anger and impulsivity) and the risk of SB during the important life-cycle transition that is adolescence. In addition, we aim to assess CM experiences and disease outcomes carefully in an approach based on a continuum of severity, dispensing of classification dichotomies that fail to reflect the complexities of reality. We hypothesize that youths with more severe CM experiences manifest higher levels of related BPD traits and are less capable of buffering the impact of negative SLEs. Thus, BPD traits and SLEs mediate the correlation between CM and risky mental states or behavior, such as NSSI or SB (see Figure 1). 

## 2. Material and Methods

### 2.1. Participants

A total of 187 children and adolescents aged 7 to 17 years participated in our multicentre study of the psychoneurobiological consequences of CM (EPI_young_stress project) [47]. Of these participants, 116 had been diagnosed with a current psychiatric disorder and 71 were healthy controls (see Table 1). Psychopathology was ascertained using the Spanish version of the Schedule for Affective Disorders and Schizophrenia for School-Age Children: Present and Lifetime Version DSM-5 (K-SADS-PL-5) [48,49]. In order to better characterize the sample, the main diagnosis was later classified into the following dimensions: attention-deficit/hyperactivity disorder, affective disorders, trauma and stress-related disorders, anxiety disorders, behavioural disorders, psychotic disorders and eating disorder (see Table 1). Youths with a current psychopathology were recruited from six child and adolescent psychiatry departments in Spain. The healthy controls were recruited at the University of Barcelona or psychiatric centres via advertisements, primary healthcare centres, schools and other community facilities. The recruitment period lasted from April 2016 to March 2020. The exclusion criteria for all participants included the diagnosis of an autism spectrum disorder, an eating disorder with BMI < 18.5, intellectual disability (IQ < 70), current drug dependence, not being fluent in Spanish, extreme premature birth (<1500 g), head injury with loss of consciousness, and severe neurological or other pathological conditions (such as epilepsy, cancer or autoimmune diseases).

Details of the assessment of the subjects have been reported elsewhere [47]. Briefly, all the participants and their parents/legal guardians were interviewed separately, face to face, by a trained psychologist or psychiatrist to obtain sociodemographic data, and their medical and psychiatric history, and to explore their CM history. 

The study was approved by the Ethical Review Board of each participating hospital and university. Families were explicitly informed of the voluntary nature of the study, their rights, and the procedures, risks and potential benefits involved. Written consent was required from all parents or legal guardians; the children provided written assent after the nature of the procedure had been fully explained.

### 2.2. Measures and Final Scores

#### 2.2.1. Childhood Maltreatment (CM)

##### CM Assessment

According to the recommendations of the National Child Traumatic Stress Network (NCTSN) [50], some key steps for conducting a comprehensive evaluation of complex trauma include the assessment for a wide range of traumatic events, and the gathering of information using a variety of techniques (clinical interviews, standardized measures, and behavioural observations) and a variety of perspectives (that of the child, caregivers, teachers, other providers, etc.). Thus, in this study, the participants and their parents/legal guardians were evaluated by trained psychologists by means of an exhaustive interview following the criteria of the instrument “Tool for assessing the severity of situations in which children are vulnerable” (TASSCV) [51] (available online in Spanish); and adolescents who were older than 12 answered self-reports such as the short version of the Childhood Trauma Questionnaire (CTQ-SF) [52] and the Childhood Experience of Care and Abuse Questionnaire (CECA-Q2) [53], while participants aged 7–11 answered an adapted ad-hoc hetero-administered questionnaire (for details see Supplementary Material in Marques-Feixa, 2021 [47]). Finally, reports from social services or teachers were reviewed by trained psychologists where applicable. After this, considering the information from the different sources, clinicians filled in a summary table, based on TASSCV criteria, regarding different forms of CM perpetrated by caregivers or other adults. Both confirmed (with clear evidence of a CM history) and suspected (if significant signs of neglect or abuse appeared during the evaluation) subjects were included. For the present report, data concerning five main types of CM were included in our analysis, which are emotional neglect, physical neglect, emotional abuse, physical abuse and sexual abuse. In our sample, 94 participants (50%) reported CM. Figure 2 shows the overlap of these CM subtypes. Additionally, clinicians assigned a severity and a frequency value to each subtype of CM, rated on a 4-point Likert scale, following TASSCV criteria. Specifically, for each subtype of CM, the severity was coded as low (1), moderate (2), severe (3) or extreme (4); while the frequency of CM exposure was coded according to whether it occurred once (1), sometimes (2), often (3) or frequently (4). 

##### CM Score

Assessing the experiences of CM on a continuum spectrum allows for a more complete and accurate understanding of the CM experiences suffered by each participant, beyond a simple presence/absence classification. Thus, a CM score was created by considering the severity and frequency of the CM: for each CM subtype, the reported severity (from 1 to 4) was multiplied by the reported frequency (from 1 to 4), to obtain a score for each subtype from 1 to 16. Finally, to obtain a global CM score, the five subtype scores were summed, thus obtaining a total CM score for each participant ranging from 0 (absence of CM) to 80 (for reports of extreme and frequent CM in all the subtypes) (see Figure 3). This CM score was included in our subsequent analysis.

#### 2.2.2. Borderline Personality Disorder (BPD) Traits

##### BPD Trait Assessment

We explored BPD traits in our sample using two different questionnaires: the Trait Emotional Intelligence Questionnaire (TEIQue) and Child Behaviour Checklist 6–18, (CBCL). On the one hand, the TEIQue test provides comprehensive coverage of facets of child personality relating to emotions (adaptability, addictive disposition, emotion expression, emotion perception, emotion regulation, low impulsivity, peer relations, self-esteem and self-motivation) [54]. More specifically, the short form of the TEIQue for children answered by parents/guardians (TEIQue-CSF) was administered to our sample, which includes 36 short statements that the participant responds to on a 5-point Likert scale (1 = completely disagree, 2 = disagree, 3 = neither agree nor disagree, 4 = agree, 5 = agree completely) [55,56]. Secondly, the CBCL is an inventory for parents of the Achenbach System of Empirically Based Assessment (ASEBA) School-Age Forms and Profiles, which assesses the competencies, behavioural and emotional problems in children and adolescents aged 6 to 18 years [57,58]. The original questionnaire contains 113 items with three response options (0 = not true, 1= somewhat or sometimes true, 2 = very true or often true). 

##### BPD Trait Score

In order to obtain BPD trait scores for all the participants, ten items from the TEIQue-CSF and eight from the CBCL were selected (see Table 2). Previously, the TEIQue-CSF scores were recoded as comparable with CBCL scores (1–2 = 0, 3 = 1, 4–5 = 2). According to the literature [59] and by reference to psychiatrists who were questioned, we chose items related to constructs that seem to be especially relevant in patients with BPD, including dysregulation and low levels of emotional control (such as affective instability, intense anger and impulsivity). To identify the relationship between items and the underlying empirical structure, we performed an exploratory factor analysis (EFA) of the principal axes with varimax rotation using SPSS. The Kaiser-Meyer-Olkin (KMO) value was 0.916, indicating the adequacy of the sampling. A parallel analysis (Monte Carlo PA software) suggested a three-factor structure for the BPD traits (see Table 2). A content analysis of the items revealed that one factor referred to the affective instability and difficulties of understanding and managing emotions. Therefore, this factor was named “emotion dysregulation”. The second factor concerned the behaviour associated with irritability, inappropriate/explosive anger and trouble controlling such anger, and was named “intense anger”. The third factor included items that relate to impulsivity of actions, and was named “impulsivity”. The three-factor composite scores were extracted (as continuous regression scores) to be used as main BPD traits. The internal consistency can be seen in Table 2. 

#### 2.2.3. Recent Stressful Life Events (SLEs)

The Life Events Inventory for Adolescents (LEIA) is a validated Spanish checklist used to screen for SLEs that occurred in youths in the last year [60]. Here, we used this instrument to assess 75 negative SLEs, including a loss or serious illness, family difficulties (financial, legal, divorce, moving address, etc.), peer problems (fights, losing a friend, breakup, etc.), academic problems (change of school, expulsion, repeating a year, etc.) and bullying or victimization, among others. The instrument showed an adequate level of reliability and good validity [60]. To avoid any interference in the main outcomes of the study, two items from the LEIA relating to one’s own mental psychopathology were excluded (“Have you suffered from any psychological or psychiatric problem?” and “Have you had any alcohol or drug-related problems?”). The LEIA “quantity score” (calculated by adding up the total SLEs (0–73)) was used in this study. 

#### 2.2.4. Risk of Suicidal Behavior (SB) 

##### Risk of SB Assessment

Since K-SADS-PL5 includes a section addressing suicide, which assesses past and current self-injurious thought and behaviour, after the interview the researchers reported information about the presence/absence of the following constructs: NSSI, suicide ideation, suicide threat, suicide plans and suicide attempts. In our sample 63 participants (33%) manifested some form of risk of SB (see Table 3 for details).

##### Risk of SB Score

The five suicide constructs previously described were considered to increase the risk of an SB score. We verified the relationships and underlying empirical structure between the five items by applying EFA with Varimax rotation via SPSS. The KMO measure of the sampling adequacy was 0.833. An inspection of the eigenvalues and screen plots suggested only one factor to be a variable. Thus, a single factor was included in main analyses as a continuous variable named “risk of SB”. 

### 2.3. Statistical Analysis

We analysed the descriptive statistics using SPSS version 26 (IBM, Chicago, IL, USA). Figure 2 was created through the use of R statistical software through the Euler diagram package. To identify relationships in our hypothesized model, we tested a path analysis model (see Figure 1) using the statistical software package EQS 6.1 [61]. Mardia’s coefficient was calculated to assess overall normality. As the model was non-normal, we used the Satorra-Bentler robust indexes [61]. To evaluate the model’s goodness-of-fit (capacity to reproduce the data), several indices were reported, including the Satorra-Bentler Chi-Square, Comparative fit index (CFI), Bollen’s fit index (IFI), McDonalad’s fit index (MFI) and the Root Mean Square Error of Approximation (RMSEA). Using standard criteria [62,63], values higher than 0.90 in CFI, IFI and MFI, and values lower than 0.08 in RMSEA, were considered as an acceptable model fit. As we needed to respecify the initial model, we used the Lagrange multiplier test (which provides information on what types of new associations could be included in the model) and the Wald test (indicating which fixed parameters or constraints might be released), according to Bentler [61]. The threshold applied for the Lagrange and Wald tests was *p* < 0.05. In addition, in order to compare the better fit of the two competing models, a Standardized Mean-square-Residual (SRMR) value was explored. When the SRMR value is found to be considerably small, the model fits the data well regardless of the other measures of fit. To compare the magnitude of each variable to predict the outcome, standardized regression coefficients were included in the arrows of path diagram (Figure 4).

## 3. Results

The descriptive data of the main variables included in the analysis can be seen in Table 3. Of the 94 participants with a history of CM, 79 (84%) have a current psychiatric disorder and 15 (16%) do not. Of the subjects without CM, 37 (40%) have a current psychiatric diagnosis and 56 (60%) do not have a psychiatric disorder. Since this is a cross-sectional study with a perspective of continue variables, the case/control differentiation was not included. Due to missing information for some of the main variables, the data obtained from nine subjects were excluded from the analysis, resulting in a final sample of 178 subjects. The attrition analysis showed no significant differences in sociodemographic factors when comparing the participants who were excluded and the subjects who were included in the path analysis.

The goodness-of-fit statistics showed that the initial model (see Figure 1) was almost overfitted (χ^2^ (3) = 2.636, *p* = 0.45, CFI = 1.00, IFI = 1.00, MFI = 1.00, RMSEA < 0.001). Hence, we decided to respecify the model. We generated an alternative model, according to the Wald test, and excluded the direct effect of CM on the risk of SB and the direct effect of intense anger and impulsivity on the risk of SB.

The alternative model fitted the data similarly to the initial model (χ^2^ (3) = 2.172, *p* = 0.53, CFI = 1.00, IFI = 1.00, MFI = 1.00, RMSEA < 0.001). Thus, we also explored the SRMR fitting statistic. According to Bentler (2006), when we compare two competing models, the one with lower SRMR should indicate the best fitting model [61]. The SRMR suggests that the best model was the respecified one, since the SRMR value of the initial model was 0.030, while that of the respecified model was 0.021. Therefore, this was used as our final model. As can be seen in Figure 4, CM had a significant positive correlation with BPD traits (emotion dysregulation (ß (0.004) = 0.345, *p* < 0.00001), intense anger (ß (0.005) = 0.263, *p* < 0.001) and impulsivity (ß (0.005) = 171, *p* = 0.009). Additionally, CM increased the risk of SLEs over the last year (ß (0.031) = 0.367, *p* < 0.00001). In turn, BPD traits are associated with higher SLE exposure (emotion dysregulation (ß (0.369) = 0.210, *p* < 0.001), intense anger (ß (0.367) = 0.259, *p* < 0.0001) and impulsivity (ß (0.333) = 0.244, *p* < 0.0001). However, only emotion dysregulation (ß (0.064) = 0.360, *p* < 0.00001) and recent SLEs (ß (0.013) = 0.419, *p* < 0.00001) showed a significant correlation with SB. This model explained 40% of the variance in the risk of SB.

## 4. Discussion

The present study examines the impact of CM on the risk of SB in a young population, exploring the mediating role of BPD traits and SLEs during the previous year. According to our results, emotional dysregulation and recent SLEs may indirectly help to explain the links between CM and the risk of SB (including NSSI, suicide threat, suicide ideation, suicide plan and suicide attempts).

On the one hand, in line with Self-Trauma Theory [28], it seems that CM (specifically, complex trauma characterized by multiple, severe and pervasive traumatic events) affects core personality domains in the early stages of life. It is hypothesised that CM may impact personality by altering the way we perceive and interpret the world around us, consequently affecting the way we respond to and manage future stressful situations [32]. From Linehan’s Biopsychosocial Model, BPD may begin with early biological vulnerability, expressed initially as impulsivity and followed by heightened emotional sensitivity, potentiated across development via environmental risk factors that give rise to more extreme emotional, behavioural, and cognitive dysregulations [29]. Specifically, as previous studies report, our findings show that CM increases the emergence of some maladaptive personality traits typically associated with BPD, such as emotion dysregulation [64], intense anger [65] and impulsivity [66]. Additionally, the emergence of other maladaptive personality dimensions has been reported in the literature, such as self-criticism [67] interpersonal difficulties [64], reduced agreeableness or openness to experience, and increased neuroticism [68]. Consequently, these types of learned adaptations could lead to a large range of dysfunctional coping behaviours over time, such as greater exposure to destructive risks (aggression, substance abuse, delinquency, risky sexual behaviour, and also SB or NSSI). Furthermore, it seems that a greater personality disturbance in those who attempt suicide seems to be associated with repeated SB [69].

In this regard, in our study, only emotion dysregulation appears as a BPD trait associated with SB. This is in accordance with a recent review of an adult population [70]. The study suggested that previous experiences of trauma should always be included as a main moderator in the association between the regulation of one’s emotions and suicide ideation or SB [70]. According to the neuroscience-based literature, CM might dysregulate neurobiological mechanisms involved in the stress response early in life, which may influence the ability to regulate emotions, resulting in reduced available internal resources to manage and respond effectively to new SLEs [47,71]. However, genetic and environmental protective factors could also lead to resilience and explain why not all people who experience such difficult starts in life experience such lifelong disability. In fact, a recent longitudinal study, based on adult psychiatric inpatients, also supports that CM was unrelated to SB, suggesting poor emotional response inhibition (a proposed behavioral marker of emotional dysregulation) as a predictor of SB [72].

Unlike other studies, the present study found intense anger and impulsivity to not be directly associated with SB [73]. Varied results have been found in past studies about this complex relationship [31]. However, the present study also agrees that emotion regulation is an importance mechanism to understand general psychopathology [46]. In fact, an inadequate emotion regulation has been associated with a wide range of both internalizing and externalizing disorders [74]. On this basis, our study adopted a transdiagnostic approach, including a wide range of mental health problems.

On the other hand, our findings also suggest that children and adolescents with higher severity/frequency of CM are more likely to experience new SLEs, thereby supporting a tendency towards revictimization [45]. Moreover, we found that the aggregation of recent SLEs are significant predictors (and indeed the main ones) of a risk of SB. Hence, the present study favours the Sensitivity–Stress Hypothesis, which states that the experience of numerous SLEs in a short period of time triggers psychopathology [75]. In addition, according to Allen et al. [25], the three BPD traits evaluated (emotion dysregulation, intense anger and impulsivity) positively correlated with exposure to further SLEs. Thus, we also offer confirmation of the Stress Generation Hypothesis of SLEs [75]. This hypothesis, first expounded by Hammen [76], proposes that some psychological characteristics (especially those related to psychopathology) could function as significant predictors of SLEs. Overall, the present study supports the existence of a self-reinforcing cycle between maladaptive traits or psychopathology and exposure to stress [75].

It is important to highlight that adolescence is a life-cycle transition and a sensitive period that often triggers overwhelming and impulsive responses to a changing environment. Therefore, youths who have been maltreated or neglected may be prone to easily triggered trauma memories that potentially bring with them great emotional pain in new circumstances—which, in the absence of sufficient emotional regulation skills, may lead the youth to engage in behaviour that reduces awareness of extreme distress, such as NSSI or SB. Interestingly, Angelakis [16] propose that while children may present high rates of suicide ideation, it is not until adolescence that SB appears. In fact, the transition to suicide attempts among adolescents with suicide ideation or NSSI may also be predicted by other variables such as cannabis and other illicit drug abuse, sleep problems and a lower extraversion score [23]. Thus, it would be interesting to evaluate subjects based on a continuum, ranging from suicide ideation to suicide attempts, during childhood and adolescence. Furthermore, considering the dose–response effect reported in the literature, a more trait-specific approach based on the multi-dimensional nature of PDs could be more useful when studying young populations [77]. Indeed, predictive personality traits in the general population, which have not yet been diagnosed or treated by mental health services, could prove useful to reducing the likelihood of risky behaviours.

### 4.1. Clinical Implications

Given that we did not find a direct association between CM and the risk of SB, the presence of mediators, such as emotion dysregulation or SLEs, provides a greater margin for intervention, thereby allowing clinicians to work on these intermediate and modifiable traits. Firstly, targeting psychotherapy towards emotion dysregulation soon after trauma (rather than prioritizing other aspects such intense anger or impulse control) would improve victims’ lives and reduce the risk of further traumatization and, ultimately, decrease the risk of SB. Secondly, it is important to consider recent SLEs in youths who have a CM history to prevent NSSI or SB [78,79]. Moreover, it is crucial to highlight that traumatized youth, although they may present other associated psychiatric disorders, often manifest disturbances in their self-organization, severe emotion dysregulation, and high levels of guilt, shame, self-harm and interpersonal problems. Thus, they require specific forms of intervention [80,81,82,83].

### 4.2. Limitations and Future Directions

Certain limitations of this study should be borne in mind. First, the present study is cross-sectional, so we cannot infer any directionality from the data. Only well-controlled longitudinal studies reveal whether the directions of the associations tested are real or spurious [84]. In addition, the study of BPD traits relied exclusively on questionaries designed to assess other variables, so we created a proxy for the main BPD features such as emotional dysregulation, intense anger and impulsivity, which did not allow for the analysis of other BPD traits. Additionally, we cannot discount the influence of genetically transmitted temperamental factors that may also influence BPD features and increase vulnerability to stress [85]. Similarly, other variables can also increase the risk of SB, such as associated mental disorders, family and socio-cultural environment or the absence of social support [73].

Furthermore, we did not include the nature of SLEs in this study. In this regard, some authors suggest that there is an association between particular stressor subtypes and suicide ideation or SB [86]; so, for future studies, it would be interesting to assess whether a particular type of SLE increases the risk of SB (e.g., SLEs with a dependent/independent nature, interpersonal or none, perpetrated by peers, or other) [17]. In addition, for distal and proximal stress factors that predispose individuals to experience problems with the regulation of their affective/cognitive or social experiences, it would be interesting to consider social learning. This may lead to self-injury over other means of self-regulation, especially in young individuals.

For future studies it would be interesting to include indirectly harmful behaviour (e.g., alcohol and substance use), since these patters commonly co-occur with directly self-injurious behaviour, and it may be useful to consider them on a continuum of self-harm behaviour. Finally, the overlap in symptomatology between different PDs, especially in younger individuals, creates a boundary problem for clinicians when making differential diagnoses. Thus, when planning appropriate and effective interventions, a more trait-specific approach based on different dimensions might be more useful than applying a dichotomous classification of PDs, widening the gap between those just above the diagnostic threshold and those at subclinical levels [87].

## 5. Conclusions

It is well understood that CM is a highly complex phenomenon that affects the individual systemically. Furthermore, it constitutes a major risk factor for the development of a large range of mental disorders, including suicide ideation, NSSI or other forms of SB. A better understanding of the mechanisms that give rise to the risk of SB, especially in children and adolescents, may have the potential to guide the development of more efficient preventative treatments and interventions. In this regard, the present study provides support for the clinical use of gathering information on specific BPD traits, such as emotion dysregulation, intense anger and impulsivity in youths with adverse childhood experiences. Our findings may suggest that, to prevent suicide ideation, NSSI and SB in young people proximally exposed to CM, it would be crucial to target interventions against emotion dysregulation and to reduce, as far as possible, their exposure to new SLEs. Additional longitudinal investigations are required to confirm this hypothesis. In any case, interventions with children and adolescents exposed to CM who manifest BPD traits requires commitment from parents, a well-coordinated medical team, and a coherent treatment schedule.

## Figures and Tables

**Figure 1 jcm-10-05293-f001:**
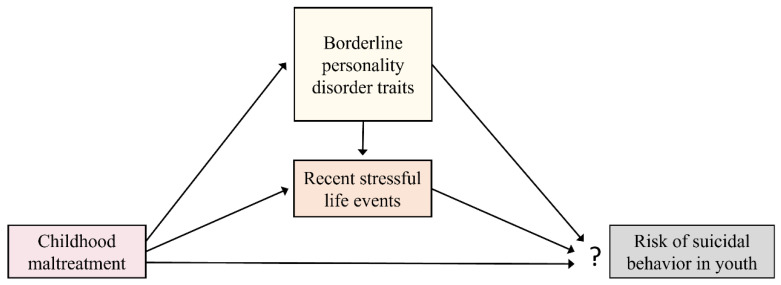
Our hypothesized model consisted of the following direct relationships: CM predicts BPD traits and is also related with recent SLEs and SB. Moreover, recent SLEs and BPD traits mediate the correlation between CM and SB. In turn, BPD traits are associated with exposure to recent SLEs and thereby also indirectly predict risk of SB.

**Figure 2 jcm-10-05293-f002:**
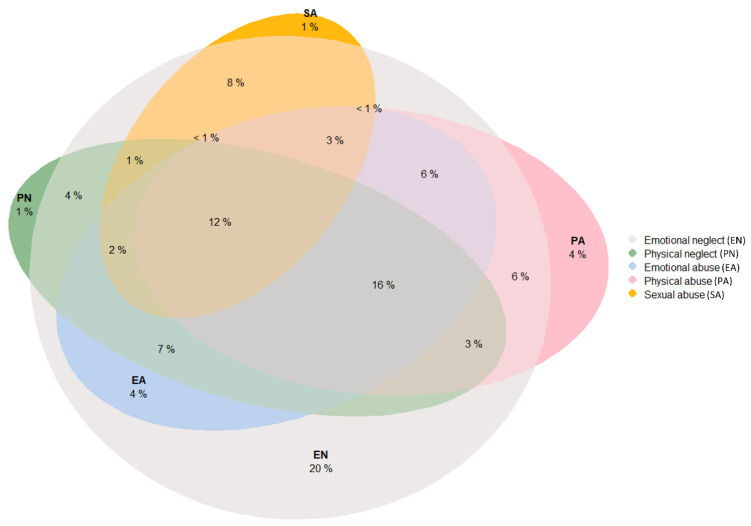
Overlap of childhood maltreatment (CM) subtypes in our sample. As can be seen, participants with a history of CM had rarely been exposed to only one type of CM. The form of CM that most often occurred in isolation was emotional neglect, accounting for 20% of the youths exposed to CM (although 94% of participants exposed to CM reported emotional neglect). Only 6% of the youth with a history of CM reported in isolation physical neglect, physical abuse or sexual abuse. Meanwhile, 74% of the individuals reported multiple forms of CM. Specifically, 16% reported four CM subtypes: emotional neglect, physical neglect, emotional abuse, and physical abuse. Furthermore, 12% of the children and adolescent exposed to CM manifested having suffered all five forms of CM, including sexual abuse.

**Figure 3 jcm-10-05293-f003:**
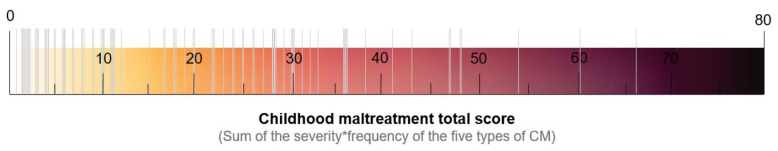
Distribution of subjects with a history of CM in the CM score (ranging from 1 to 80), considering the severity and frequency of each form of CM (emotional neglect, physical neglect, emotional abuse, physical abuse and sexual abuse).

**Figure 4 jcm-10-05293-f004:**
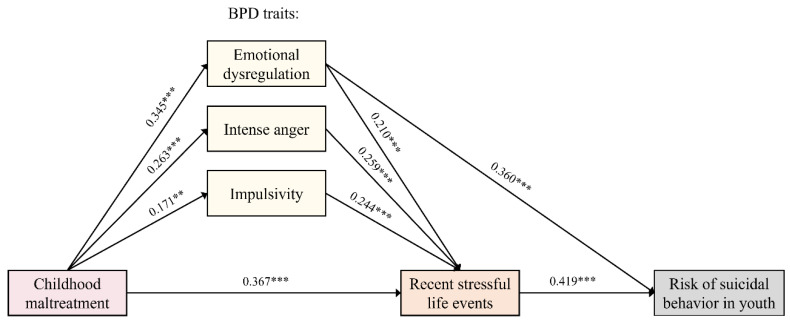
Final model obtained by path analysis. Our results support the hypothesis that youths with a more severe/frequent history of CM present a higher risk to recent SLEs. Furthermore, we found that youths with greater CM exposure had more BPD traits (emotion dysregulation, intense anger and impulsivity), which, in turn, directly predict higher incidence of recent SLEs. Nevertheless, SB was only correlated with emotion dysregulation and recent SLEs, but not with CM or the other BPD traits (impulsivity or intense anger) as we hypothesized. *p* values: ** *p* ≤ 0.01, and *** *p* ≤ 0.001.

**Table 1 jcm-10-05293-t001:** Demographic and clinical descriptive data of our sample (*n* = 187).

Variables		Value
**Age**—mean (Sd) [range]		13.62 (2.59) [7–17]
**Sex**	Female—*n* (%)	108 (58%)
	Male—*n* (%)	79 (42%)
**Ethnicity**	European—*n* (%)	154 (82%)
	Others ^a^—*n* (%)	33 (18%)
**Socioeconomic status (SES)** ^b^—mean (Sd) [range]		40 (18) [8–66]
**Current psychiatric diagnosis status**	Without current psychiatric diagnosis—*n* (%)	71 (38%)
	With current psychiatric diagnosis—*n* (%)	116 (62%)
	**Primary psychiatric diagnosis dimensions:**	
	ADHD ^c^—*n* (%)	30 (26%)
	Affective disorders—*n* (%)	29 (25%)
	Trauma and stress-related disorders—*n* (%)	19 (16%)
	Anxiety disorders—*n* (%)	15 (13%)
	Behavioural disorders—*n* (%)	13 (11%)
	Psychotic disorders—*n* (%)	7 (6%)
	Eating disorders—*n* (%)	3 (3%)

^a^ Other ethnicities were: Latin American (66%), Maghrebin (16%), sub-Saharan (9%), and others (9%). ^b^ SES is based on the Hollingshead Four-Factor Index of socioeconomic status (Hollingshead, 1975); higher scores reflect higher SES. ^c^ ADHD, attention-deficit/hyperactivity disorder.

**Table 2 jcm-10-05293-t002:** Borderline personality trait scores.

Borderline Personality Disorder Traits	Item	Original Test (Item)	Loading Factor	Cronbach Alpha
Emotion Dysregulation	She/he can’t find the right words to tell others how she/he feels	TEIQue-CSF (29)	0.720	0.804
	She/he is often confused about the way she/he feels	TEIQue-CSF (33)	0.711	
	She/he feels great about her/himself	TEIQue-CSF (4)	−0.641	
	Refuses to talk	CBCL (65)	0.625	
	It’s easy for her/him to understand how she/he feels	TEIQue-CSF (16)	−0.618	
	Often, she/he is not happy with her/himself	TEIQue-CSF (12)	0.578	
	She/he is not good at controlling the way she/he feels	TEIQue-CSF (27)	0.530	
Intense Anger	Stubborn, sullen or irritable	CBCL (86)	0.845	0.915
	Temper tantrums or hot temper	CBCL (95)	0.790	
	Sulks a lot	CBCL (88)	0.758	
	Whining	CBCL (109)	0.752	
	Sudden changes in mood or feelings	CBCL (87)	0.710	
	Argues a lot	CBCL (3)	0.667	
	She/he often feels angry	TEIQue-CSF (5)	0.587	
Impulsivity	She/he thinks very carefully before she/he does anything	TEIQue-CSF (26)	−0.846	0.853
	Many times, she/he doesn’t think before she/he does something	TEIQue-CSF (13)	0.806	
	Impulsive or acts without thinking	CBCL (41)	0.716	
	Usually, she/he thinks very carefully before she/he talks	TEIQue-CSF (36)	−0.715	

Note: CBCL: Child Behavior Checklist 6-18. TEIQue-CSF: Short form of the Trait Emotional Intelligence Questionnaire for children answered by parents/guardians.

**Table 3 jcm-10-05293-t003:** Main variables studied.

Variable			Value
**CM**	Absence (*n*, %)		93 (50%)
	Presence (*n*, %)		94 (50%)
	**Types of CM:**	Emotional neglect (*n*, %)	84 (44%)
		Physical neglect (*n*, %)	43 (23%)
		Emotional abuse (*n*, %)	49 (26%)
		Physical abuse (*n*, %)	50 (27%)
		Sexual abuse (*n*, %)	28 (15%)
	**CM total score**—mean (Sd) [range]		9.50 (14.26) [0—66]
**Recent SLEs**	**Total SLEs**—mean (Sd) [range]		6.60 (5.77) [0—25]
**BPD trait score**	Emotion dysregulation—mean (Sd) [range]		0.00 (1.0) [−1.96–2.91]
	Intense Anger mean (Sd) [range]		0.00 (1.0) [−1.89–2.26]
	Impulsivity mean (Sd) [range]		0.00 (1.0) [−2.28–2.15]
**SBs**	Absence (*n*, %)		124 (67%)
	Presence (*n*, %)		63 (33%)
	**Subtypes of** **SBs:**	Non-suicidal self-injury (NSSI) (*n*, %)	48 (26%)
		Suicide ideation (*n*, %)	45 (24%)
		Suicide threat (*n*, %)	33 (18%)
		Suicide plans (*n*, %)	18 (10%)
		Suicide attempt (*n*, %)	30 (16%)
	**Risk of****SB score**—mean (Sd) [range]		0.0 (1.0) [−0.58–2.63]

Note: BPD: borderline personality disorder. CM: childhood maltreatment. SBs: suicidal behaviours. SLEs: stressful life events.

## Data Availability

The data presented in this study are available on request from the corresponding author. The data are not publicly available due to restrictions in privacy and ethical.

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
