# Peer review of "Risk of Suicidal Behavior in Children and Adolescents Exposed to Maltreatment: The Mediating Role of Borderline Personality Traits and Recent Stressful Life Events"

_jcm, 2021, doi:10.3390/jcm10225293_

Round 1
Reviewer 1 Report
General Comments:
A worthwhile study that examines the contributions of childhood maltreatment, BPD traits and recent stressors to risk of suicide in children and adolescents. The complex interplay among these experiential and potentially epigenetic factors is evaluated using structural equation modelling. The finding that emotion dysregulation and recent stressors were the best predictors of suicide leads to their conclusion that interventions to address emotional dysregulation and protection from new stressful life events could reduce suicidal behaviour.
The choice of participants for this study (and lack of explicit use of a comparator group of young people who did not have early life trauma) may unfortunately create significant bias in attributing predictive relationships among these variables. The experience of childhood trauma does not necessarily lead to these outcomes. This is despite the evidence that a high proportion of people diagnosed with BPD have experienced trauma, often in early life. The genetic vulnerability to developing BPD is not factored in either.
The text seems rather wordy at times. Try to reduce the inclusion of multiple ideas within single paragraphs (Discussion) as this dilutes the argument. Review carefully to improve English language, grammar, and expression.
There is an overuse of non-standard abbreviations; this is distracting to the reader.
Specific Comments:
Abstract:
It is unclear what is meant by ‘EQS’
Suggestion for different phrasing ‘…more severe/frequent CM have more prominent BPD traits,…’
Introduction:
Consider changing ‘plastic’ to ‘neuroplastic’
The first paragraph paints a grim picture. It is worth noting that there is a role for resilience and other protective factors; not all people who experience such difficult starts in life experience such lifelong disability. There should also be some mention that suicidal and self-injurious behaviour is not necessarily preceded by childhood maltreatment.
Second paragraph: ‘SA’ needs to be defined on first mention.
Replace ‘might be emerged’ with ‘might emerge’
Replace ‘Specially’ with ‘Especially’
Fifth paragraph: ‘PD’ needs to be defined on first mention.
When referring to ‘risky personality traits’ are you referring to ‘risky behaviour’? The way it is described makes it sound temporary (i.e. most grow out of it)
Methods: The use of the expression ‘...CM exercised by caregivers or other adults.’ is unusual. Might ‘perpetrated’ be clearer?
Figure 2 is visually a bit difficult to understand although the legend does clarify this considerably.
Figure 3 is very effective in communicating the range of CM severity scores.
The measurement of BPD traits uses methodology which adapted questions from other instruments thought to reflect some aspects of BPD. While this has good face validity, did the authors consider using the Borderline Personality Features Scale for Children (BPFS-C) and its associated parent report version?
Was evidence of childhood maltreatment found in any of the 71 children without a psychiatric diagnosis? The results for these participants don’t appear to feature in the results of this study.
There are quite a few spelling errors in section 2.3.
With respect to the model that was chosen, the initial model (which is not specified) was abandoned due to ‘overfitting’. However the statistics for the modified model appear almost identical. It is unclear what the difference is between these ‘goodness of fit’ statistics. How might the authors explain why excluding the direct effect of CM on risk of suicide may have made a difference to their analysis? Were other variables (or links between certain variables) also investigated to see how they affected the model?
The legend for Figure 4 needs to include what is meant by ‘***’
At first glance, the highly significant association across all chosen variables seems to imply some redundancy and/or circularity among the variables chosen for study. The high degree of inter-relatedness makes it difficult to determine the relative importance of any particular variable. Were there any significant associations outside of those depicted here (e.g. between impulsivity and risk of suicide)? Perhaps I misunderstand the analysis by asking these questions.
Discussion:
Third paragraph: It is unclear whether the authors are referring to their own study results in the following sentence ‘However, unlike other studies, intensive anger and impulsivity have not been shown as direct factors of risk of suicide (69).’ It appears that they might be but this needs to be clarified in light of the subsequent sentences. The argument for the inter-relatedness of anger, impulsivity and emotion dysregulation is confusing since these are considered separately in the model and seem to be differentiable from emotion dysregulation when it comes to suicidal behaviour.
The inclusion of information implicating the HPA axis seems out of place in this paragraph and perhaps superfluous to this study as it has not been mentioned previously.
Although it is pointed out in the Limitations that directionality cannot be inferred from cross-sectional data, the expression ‘direct’ is used repeatedly in the paper when referring to associations among the variables.
Equating CM to ‘violence’ in the last sentence seems misplaced.
Reviewer 2 Report
Title: Risk of suicide in children and adolescents exposed to maltreatment: the mediating role of borderline personality traits and recent stressful life events
Summary: Thank you for inviting me to review this fascinating and generally well-written manuscript. The authors of this study evaluate hypothesized dispositional and environmental mechanisms (i.e., stressful life events; SLEs) underlying heightened suicide risk among maltreated children and adolescents, grounded in an established theoretical framework (i.e., Myers et al.’s “Self-Trauma Model”) for multi-final self-damaging outcomes associated with repeated maltreatment exposure (c.f., “complex trauma”) in youth. The authors hypothesize moderating/mediating (indirect) effects of borderline personality disorder (BPD) traits and recent negative SLEs in the cross-sectional association between maltreatment history and deliberate self-harm – both nonsuicidal self-injury (NSSI) and suicidal behaviors (SBs). The final model obtained via path analysis partially supports this hypothesis, indicating specifically that emotional dysregulation and SLEs help account for the expected relationship between childhood maltreatment history and “suicide risk”, a composite variable derived from interviews assessing the presence or absence of NSSI, suicidal ideation, and SB histories. Childhood maltreatment itself did not demonstrate a direct effect on suicide risk, after accounting for BPD traits in the model (particularly emotional dysregulation), consistent with some recent literature. Taken together, these findings align with prior research implicating emotional dysregulation as a core dimension of myriad psychopathologies beyond BPD, and further comport with a large body of work demonstrating detrimental effects of early life maltreatment on the development of cognitive capacities necessary for self-regulation, which often persist into adulthood, manifesting as various psychiatric syndromes and/or self-injurious behaviors. The reviewed study accordingly advances our understanding of suicide risk in youth and motivates future prospective work to confirm observed indirect effects as true “mediators”. I have noted several minor items below (separated by article section) that would improve this submission; this manuscript has strong potential for publication in the Journal of Clinical Medicine, should the authors’ revisions adequately address each point.
Relative strengths:
- Overall clarity (although some grammatical/syntactical proofreading is needed)
- Succinct but sufficiently comprehensive review of relevant empirical literature and theoretical work in the Introduction, which nicely sets up primary hypotheses
- Focus on a spectrum of psychopathological behavior rather than discrete psychiatric disorders
- Appreciable sample size with a wide age range that seems fairly representative of the region from which participants were recruited, with diverse transdiagnostic psychiatric diagnoses
- Use of multimodal assessment of maltreatment combining interviews and self-/informant-report
- Figure 2 is a very helpful visualization
Introduction:
- The Introduction provides a generally cogent review of the empirical literature that adequately motivates the study aims and hypotheses. In terms of conceptual work, however, there are a few notable omissions that (if incorporated) could broaden the implications of this research. For example, the authors’ heavy reliance on the “Self-Trauma Model” seems limiting, given that the studied constructs and proposed hypotheses align with a variety of theories, e.g., Linehan’s Biopsychosocial Model of BPD, the integrated framework linking NSSI and suicide put forth by Hamza, Willoughby, and Stewart (2013), and perhaps most importantly, the functional theory of the p factor proposed by Carver, Johnson, and Timpano (2017) – which speaks both to emotional dysregulation and I thus strongly suggest that the authors consider acknowledging how their study fits within (and/or could be informed by) these closely related theoretical frameworks. It would be appropriate and illuminating to outline conceptual relationships between similar constructs invoked in other relevant prominent theories (e.g., aggression vs. emotion-related impulsivity vs. emotional dysregulation vs. impulsivity), ideally integrating a bit more evidence from affective neuroscience, experimental psychopathology, and/or neuropsychological research – especially given (1) the sensitive neurodevelopmental window on which this study is focused and (2) the wealth of extant empirical work examining the development of cool and hot executive functions in childhood and adolescence (which should be addressed either in the Introduction or Discussion). In this vein, I encourage the authors refer to our recent review on neurocognitive factors in adult suicide (Allen, Bozzay, & Edenbaum, 2019), which discusses emotional dysregulation as a “multi-final common pathway” through which disparate diatheses (including childhood maltreatment) operate to influence varied adverse clinical outcomes – particularly NSSI/SBs.
Methods:
- I have no major methodological concerns regarding study design or analyses.
Results:
- Findings are largely consistent with hypothesized effects and reported in a generally clear manner, with useful visual aid provided by the included figures.
Discussion:
- The second sentence in this section states, “According to our results, emotion dysregulation and recent SLEs should be considered as full mediators [emphasis added] in the association between CM and risk of suicide…” I do not categorically take issue with using the term “mediator” in cross-sectional analyses (as many colleagues do), but this wording should be tempered to better match the study results, e.g., “emotional dysregulation and recent SLEs may indirectly help explain links between CM and suicide risk.” I further suggest additional caution throughout the manuscript to avoid using language implying causality.
- On page 14, the authors write: “unlike other studies, intense anger and impulsivity have not been shown as direct factors of risk of suicide.” A considerable body of literature contradicts this assertion; see the review listed above for a selection of such references (Allen et al., 2019). The authors should therefore either note that findings in this area are mixed or clarify if they are referring to null effects of those constructs (on suicide risk) as measured by particular means or operationalized in a certain manner, e.g., self-report vs. EMA vs. behavioral tasks.
- Finally, I invite the authors to consider mentioning how their findings comport with our recent longitudinal work examining relations among childhood maltreatment, emotional response inhibition (a proposed behavioral marker of emotional dysregulation), and suicide in psychiatric patients (Allen et al., 2021). The implications of these studies seem highly relevant to one another and warrant a brief consideration of how parallel advances in the study of emotional dysregulation and suicide in youth and adult populations might reveal similar or distinct mechanistic trajectories from the same diathesis (e.g., maltreatment history).
